# Predicting the efficiency of oxygen-evolving electrolysis on the Moon and Mars

Bethany A. Lomax [1,2✉], Gunter H. Just [3], Patrick J. McHugh [1], Paul K. Broadley [3], Gregory C. Hutchings [3], Paul A. Burke [4], Matthew J. Roy [3], Katharine L. Smith [3] & Mark D. Symes [1✉]

Establishing a permanent human presence on the Moon or Mars requires a secure supply of oxygen for life support and refueling. The electrolysis of water has attracted significant attention in this regard as water-ice may exist on both the Moon and Mars. However, to date there has been no study examining how the lower gravitational fields on the Moon and Mars might affect gas-evolving electrolysis when compared to terrestrial conditions. Herein we provide experimental data on the effects of gravitational fields on water electrolysis from 0.166 g (lunar gravity) to 8 g (eight times the Earth's gravity) and show that electrolytic oxygen production is reduced by around 11% under lunar gravity with our system compared to operation at 1 g. Moreover, our results indicate that electrolytic data collected using less resource-intensive ground-based experiments at elevated gravity (>1 g) may be extrapolated to gravitational levels below 1 g.

[1] WestCHEM, School of Chemistry, University of Glasgow, Glasgow G12 8QQ, UK. [2] ESA - European Space Research and Technology Centre, Keplerlaan, Noordwijk 2201 AZ, Netherlands. [3] Department of Mechanical, Aerospace and Civil Engineering, University of Manchester, Oxford Road, Manchester M13 9PL, United Kingdom. [4] Space Exploration Sector, Johns Hopkins University Applied Physics Laboratory, Laurel, MD, USA. ✉email: beth.lomax@esa.int; mark.symes@glasgow.ac.uk

Electrolytic oxygen production (be this by the electrolysis of water[1–5], or by the electrolysis of regolith in molten salts or oxides[4,6–8]) will be critical for sustainable habitation of the Moon and Mars. This subject may have seemed of only academic interest a few short years ago, but recent commitments by national agencies and commercial players to return astronauts to the lunar surface and to establish a permanent human presence on the Moon provide an urgent imperative to develop new approaches for supporting life on the Moon from resources found in-situ.

The three-phase interfacial phenomena that determine the behavior of oxygen bubbles as they form at the electrode surface during electrolysis of water are strongly dependent on gravity[9]. The average gravitational acceleration on Earth is 9.807 m/s² or 1 g; gravity on Mars and the Moon is approximately 1/3 and 1/6 of the gravity on Earth, respectively. As the bubble evolution behavior directly influences the electrochemical efficiency of oxygen production, the impact of gravity on electrolyzers operating on the Moon or Mars needs to be better understood[10]. There are multiple examples of studies investigating electrolytic gas production under very low gravity (microgravity) conditions in drop towers ($10^{-6}$ g)[2,11–16], and during parabolic flight ($10^{-2}$ g)[17–20]. However, it is far from obvious as to whether such low gravity regimes can be extrapolated to the prevailing gravitational conditions on the Moon or Mars. Moreover, there has yet to be an experimental investigation of electrolytic oxygen production at lunar or Martian gravity. This knowledge-gap forms a considerable barrier to the design of electrochemical systems for oxygen generation on the Moon and Mars.

Bubbles on an electrode surface typically evolve via four phases: nucleation, growth, coalescence, and detachment[9]. A balance of various forces determines if a bubble remains attached to the electrode surface or departs into the electrolyte; the buoyant force acting on the bubble is proportional to the gravitational force and acts to drive the bubble off the electrode surface and upwards through the electrolyte against the direction of gravity[9,21]. Under microgravity, where the buoyancy is negligible, the interfacial tension force that holds a bubble to a surface dominates, and as such, bubbles grow larger[15]. A thick bubble froth layer around an electrode in microgravity has been shown to increase the overpotential for electrolysis by blocking the electrode surface, increasing ohmic resistance, and hindering the transfer of reactants to (and products away from) the electrode surface[2,12–14,16]. This process is visualized for a lower-gravity vs. a higher-gravity case in Fig. 1. The overpotential of an electrochemical half reaction ($\eta$) can be divided into the contribution from activation overpotential ($\eta_a$), concentration overpotential ($\eta_{conc}$), and ohmic overpotential ($\eta_{ohm}$). Gas bubbles attached to an electrode surface act to increase $\eta_a$ due to a decrease in the effective electrocatalytic area, increase $\eta_{ohm}$ by limiting current flow to the surface and in the bubble diffusion zone, and decrease $\eta_{conc}$ by creating reservoirs for dissolved gas products[10]. As such, a clear relationship between gravity-induced changes in bubble behavior at electrodes and the overpotential of an electrochemical system has been established. Bubble behavior has also been shown to be heavily dependent on electrolyte composition and pH[2,12,13], as well as electrode surface properties and/or modification[15,16,22–24].

The same gravity-dependent interfacial phenomena that reduce electrolysis efficiency in microgravity have been shown to increase efficiency in hypergravity electrolysis (>1 g). Hypergravity experimental conditions can be achieved by using a centrifuge to generate centripetal acceleration, giving g-levels equivalent to many times ambient gravitation. Under hypergravity conditions, the critical radius for bubble detachment is reduced, and greater buoyancy allows for rapid expulsion of gas from the electrode surface. This, in turn, exposes a greater

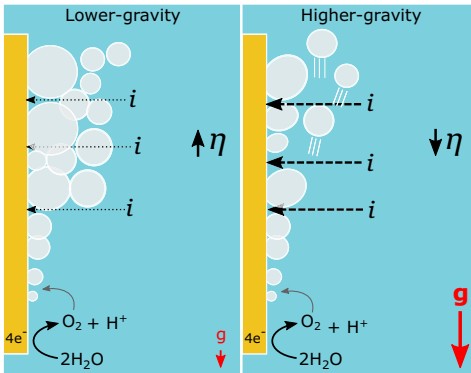

**Fig. 1 Influence of bubbles on overpotential in different gravity levels.**
A comparison of a lower-gravity vs. a higher-gravity scenario where electrolytic production of oxygen via the electrolysis of water leads to the nucleation, growth, and coalescence of bubbles at the surface of the electrode. In the lower-gravity case, bubbles will adhere more to the surface, increasing ohmic resistance and increasing overpotential compared to the higher-gravity system, where bubbles will detach more rapidly from the electrode, reducing resistance in the system and lowering overpotential. The gravity-levels have been left arbitrary as this relationship applies to any comparison of different gravity environments. i = current flow; g = gravity; $\eta$ = overpotential.

effective electrode area, decreasing overpotential, and increasing efficiency[25–28]. The primary focus of hypergravity work to date has been examination of process intensification with the application of very high rotational speeds; the large gravity ranges investigated (often >100 g) make extrapolation below 1 g less reliable[29]. Additionally, as quantitative assessment of the influence of gravity is highly dependent on the particular electrochemical system under test, inter-study comparison is challenging. The aim of the present study was therefore to use a simple aqueous electrochemical system to investigate the efficiency of oxygen-evolving electrolysis in lunar and Martian gravity conditions, and to provide experimental verification as to whether or not ground-based hypergravity research platforms could be used to estimate expected efficiency losses under reduced-gravity conditions, such as those found on the Moon or Mars. If such data can be collected without the need for parabolic flights, this would be a major advantage for research into electrolysis under reduced-gravity conditions as parabolic flights are expensive, not widely available and impose restrictions on what experimental conditions can be studied (e.g., the production of hydrogen must be strictly controlled, and the period of reduced-gravity is typically no more than 20 s at a time). Ground-based centrifuge experiments on the other hand can run for much longer durations, with more flexible operating requirements and such facilities are generally cheaper and more accessible than parabolic flights. To undertake this study, altered-gravity experiments were carried out between microgravity (~0.01 g) and 8 g. Investigation of the same electrochemical system across all gravity-levels was critical to allow comparison of the data collected above and below 1 g.

We show herein that the same general trend in the efficiency of oxygen-evolving electrolysis exists above and below 1 g, whereby lower gravity conditions lead to lower electrolysis efficiencies relative to higher gravity conditions. A logarithmic trend similar to that which can be extrapolated from hypergravity data is observed in the region between 0.166 g and 1 g, suggesting that hypergravity research platforms can indeed provide data that can be used to predict the efficiency of electrochemical systems operating in reduced-gravity. Based on the data obtained, the

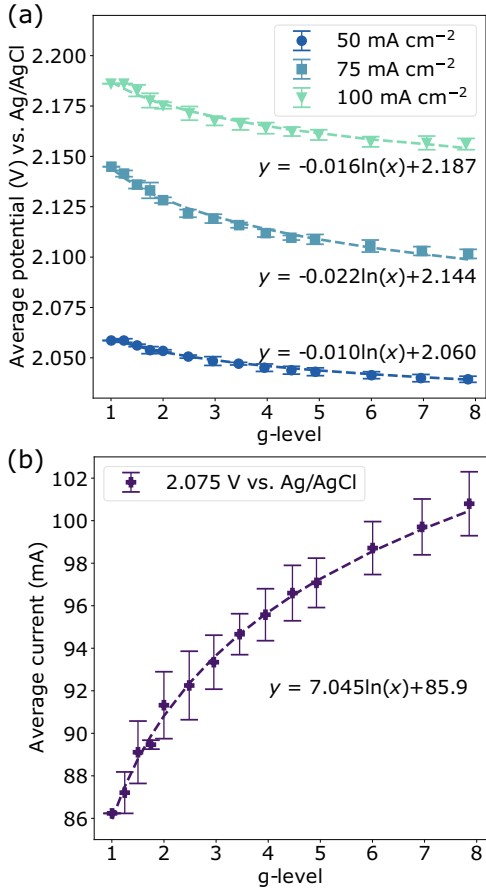

**Fig. 2 Hypergravity gas-evolving electrolysis (1–8 g).** The average potential during galvanostatic experiments (**a**) and average current during potentiostatic experiments (**b**) with hypergravity conditions achieved using a centrifuge. Error bars represent the standard error of the mean. To remove the influence of cell-specific shifted baselines, the average % change from 1g for each g-level was applied to the mean 1g value. $R^2$ values are >0.98 for all the fitted trends shown. Source data are provided as a source data file.

electrochemical cell in the present study would produce 11% less oxygen if the increased overpotential requirement was not accounted for when operating potentiostatically, compared with a cell operating with equivalent parameters on Earth. Alternatively, the additional power required to produce the equivalent amount of oxygen by maintaining a current density of 100 mA cm$^{-2}$ would be modest, with approximately 1% additional power required under lunar conditions. This study highlights the relationship between power consumption, product yield, and gravity, and suggests that appropriately designed ground-based hypergravity systems can be used to determine the ideal operating conditions for a given system in low gravity, potentially negating the need for costly and complex parabolic flight experiments.

## Results and discussion

**Hypergravity oxygen-evolving electrolysis.** To compare with the data collected during parabolic flight, oxygen-evolving electrolysis experiments were carried out under hypergravity conditions from 1 to 8 g using a ground-based centrifuge ($r = 25$ cm). The average potential as a function of g-level is shown for all galvanostatic experiments in Fig. 2a. All the data follow a logarithmic trend, with a lower potential required to meet a fixed current at 8 g relative to 1 g. Comparison of the potentiostatic data (Fig. 2b) shows that they follow a similar logarithmic trend, where the

average current in a given experiment increases as gravity increases. These data represent the average of two data sets taken for all parameters, which followed an equivalent trend in all cases. Between 1 and 8 g, the change in required potential is 0.9–2% across all galvanostatic data sets, whereas the current at 8 g increases by ~17% relative to the current at 1 g. This demonstrates that a relatively small change in the anode overpotential can have a significant impact on the current when running this system potentiostatically.

Overall, the general trends identified are in line with previous literature, whereby increased gravity and greater buoyant forces enhance bubble removal from the electrode surface and improve the performance of the electrochemical cell[9,26–28]. Previous work has reported that the relationship between the cell voltage ($E$) and gravity ($g$) for a given current density for gas-evolving electrolysis in a hypergravity field can be expressed by Eq. (1)[26,30]:

$$E_g = \beta \log(g) + E_1 \tag{1}$$

where $E_g$ is the cell voltage at a given gravity, $\beta$ is the rate of change, and $E_1$ is the potential at 1 g. Wang et al.[26] investigated current densities from 100 to 800 mA cm$^{-2}$; the relationship at 100 mA cm$^{-2}$ was found to be:

$$E_g = -0.05 \log(g) + 1.43 \tag{2}$$

Conversion of the slope found for 100 mA cm$^{-2}$ in the present work to $\log_{10}$ for the centrifuge data gives Eq. 3.

$$E_g = -0.04 \log(g) + 2.19 \tag{3}$$

Thus, the data obtained at hypergravity on the centrifuge at 100 mA cm$^{-2}$ follow a similar trend to that previously suggested by Wang[26]. We note that this previous study investigated water electrolysis in basic conditions with a focus on hydrogen gas production, whereas the current work looks at oxygen production under acidic conditions; differences in the finer detail of these trends is, therefore, to be expected, but nevertheless the overall trends are in good agreement.

**Reduced-gravity oxygen-evolving electrolysis.** The efficiency of oxygen-evolving electrolysis in reduced-gravity conditions was investigated using four electrolysis cells on a centrifuge operating during microgravity parabolic flights. The same data sets were collected as in the hypergravity study; two galvanostatic data sets (50 and 100 mA cm$^{-2}$) were also repeated in descending g-level order to account for any changes in the electrolyte over the course of a flight (see Supplementary Fig. 11 and Supplementary Notes 3 and 4). Data were collected at nine reduced-gravity levels, including lunar gravity (0.166 g) and Martian gravity (0.376 g). Examination of reduced-gravity levels below 0.166 g was attempted, however, the centrifugal acceleration in these experiments was not sufficient to reorient the electrolyte and overcome the surface tension between the electrolyte and the cell walls, resulting in electrolyte being partially on the roof of the cell at g-levels below 0.166 (see Supplementary Fig. 12). This resulted in current loss via the pressure sensor ground connection, rendering data collected below 0.166 g unreliable. Some experiments between 0.166 and 0.5 g were also impacted by this issue and were therefore removed from subsequent analysis. Comparative microgravity data were collected with two stationary upright cells adjacent to the centrifuge (i.e. not undergoing rotation to impose a particular g-level).

Figure 3 shows the average potential as a function of gravity across all reduced-gravity galvanostatic data sets (a) and the average potential as a function of gravity for all potentiostatic data sets (b). The average of all cells operating in microgravity is also shown. While the electrochemical system was the same across all

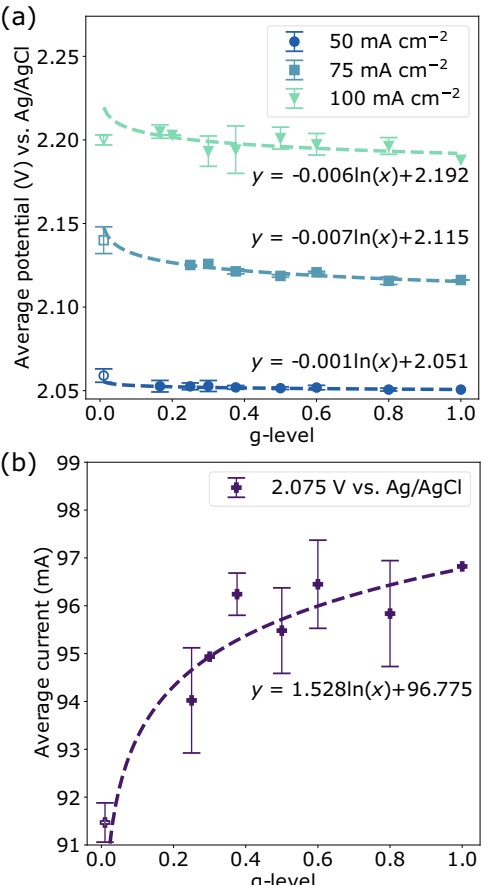

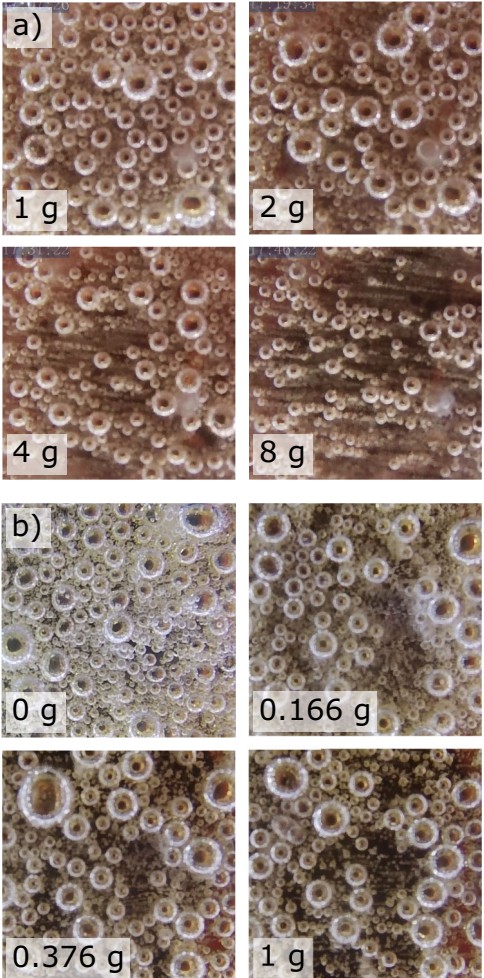

**Fig. 3 Reduced gravity gas-evolving electrolysis (0–1 g).** The average potential or current of all reduced-gravity electrolysis experiments under galvanostatic (**a**) and potentiostatic (**b**) control, respectively (data collected with centrifuge (filled) or stationary (unfilled) cells). Error bars represent the standard error of the mean. To remove the influence of cell-specific shifted baselines, the average % change from 1 g for each g-level was applied to the mean 1 g value; hence 1 g data points and g-levels where only a single data point was collected are not shown with error bars. Trends shown are calculated based on centrifuge data only and have an $R^2$ value of 0.848, 0.868, and 0.496 for 50, 75, and 100 mA cm$^{-2}$ respectively, and 0.651 for 2.075 V. Source data are provided as a source data file.

**Fig. 4 Bubble behavior in various gravity levels.** A comparison of 1 cm$^2$ frames of the bubbles on the electrode surface ($t = 14$ s) collected at 50 mA cm$^{-2}$ in: **a** laboratory conditions using a centrifuge to generate 1–8 g, and **b** parabolic flight using a centrifuge to generate 0.166–1 g.

cells, every data set had a slightly different baseline, likely due to the custom-made electrodes and possible variation in reference electrodes. Additionally, each data set contained a different number of g-levels based on the quantity of data in a given set impacted by the aforementioned electrolyte reorientation issue at low g-levels. Consequently, taking a direct average at each g-level did not fairly represent the data. To assess the mean influence of gravity for each parameter, the percentage change relative to the 1 g data point was calculated for every individual data set. This percentage change was then averaged for each g-level and applied to the mean value obtained at 1 g to give the average potential/current as a function of gravity shown in Fig. 3. The reduced-gravity data appears to follow the same general logarithmic trend as predicted from the hypergravity results, where electrolysis becomes less efficient as gravity is reduced. However, the relationship with gravity is less pronounced and the larger error in all the parabolic flight data averages indicates that this result may require further validation in future experiments.

By necessity, the comparison of microgravity and 1 g had to be made between different cells; it was not possible to collect 1 g controls for the stationary cells as every parabola provided 10$^{-2}$ g

and we were not permitted to operate these stationary cells outside of a parabolic maneuver. Hence, the average of 1 g data collected in cells on the centrifuge was used to provide a 1 g benchmark. Therefore, there is potential for inherent error within this comparison as each cell has a slightly different baseline as discussed previously. Microgravity data was in reasonable agreement with the trend line extrapolated from the reduced-gravity centrifuge data in all cases except for a current density of 100 mA cm$^{-2}$; baseline shift likely accounts for this discrepancy.

**Bubble behavior in altered-gravity electrolysis.** To better understand the behavior of the bubbles at the electrode surface and how they are influenced by gravity, video footage was recorded of the electrode face and side. A selection of front-angle frames at four hypergravity levels is shown in Fig. 4a, where the size of bubbles on the surface appears to decrease with increasing g-level. Frames showing 0–18 s at each g-level examined between 1 and 8 g can be seen in Supplementary Fig. 13.

Comparative examination of the bubble behavior between 0 and 18 s under different gravity levels below 1 g (Supplementary Fig. 14) showed that the formation of a froth layer of small bubbles at the electrode surface appears to be more prevalent at the lower gravity levels. A close-up comparison of four frames from experiments at 0, 0.166, 0.376, and 1 g is shown in Fig. 4b.

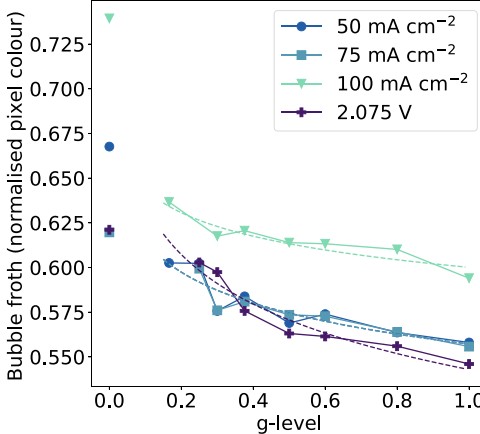

**Fig. 5 Bubble froth coverage analysis.** The average bubble froth coverage fraction as a function of gravity for each parameter based on normalized average pixel color (0 = black; 1 = white). Logarithmic trends are fitted between 0.166 and 1 g; $R^2$ values for all trend lines shown are >0.828. Average coverage values for all stationary cells in microgravity are shown as single points but are not included in the fit as they are independent from data sets collected with the centrifuge. Source data are provided as a source data file.

The average size of the larger bubbles did not appear to change in an appreciable way between 0.166 and 1 g; however, an increase in small-diameter fizz can be seen between the larger bubbles at the lower g-levels. Compared with 1 g and partial-g, the maximum bubble size in microgravity does not appear to increase, but significantly more medium and small-sized bubbles appear to be attached to the electrode surface. The observation of increased froth layer at the electrode surface under microgravity conditions is consistent with previous work[12,13,18]. The comparison between the microgravity and partial-gravity bubbles indicates that bubble behavior changes more dramatically between 0.166 g and microgravity than it appears to change between 1 g and 0.166 g, consistent with the identified logarithmic relationship between gravity and electrolysis efficiency.

Quantitative analysis of average bubble size and electrode coverage was not possible due to the convoluted peripheries of the bubbles (their fizziness). However, the bubble froth coverage on the electrode in reduced-gravity was approximated by a comparative analysis of the color of each frame following conversion to grayscale. Figure 5 shows the results of this analysis, where a y-axis value of 1 is equal to white and indicates more bubble froth, and a y-axis value of 0 is equal to black and indicates that more of the electrode is exposed. While larger bubbles would increase the coverage of the electrode surface, the dark interior of these bubbles would lower the froth coverage value as defined by pixel color. The observation that the size of the larger bubbles does not change significantly from 0.166 to 1 g indicates that pixel color is a fair approximation of electrode coverage. All reduced-gravity data sets were averaged; the fitted trend for each parameter is shown to follow the same logarithmic trend as that seen in the electrolysis data. The froth coverage is also shown for microgravity cells. In all cases, the greater froth coverage observed in the visual analysis of the footage translates to a higher normalized froth coverage value based on pixel color values. Bubble production is proportional to current flow, thus the pixel color averages would be expected to increase in the order 50 < 75 < 100 mA cm⁻². The baseline value for each cell is highly dependent on the cell lighting, and this explains why the 50 mA cm⁻² data largely overlaps with the 75 mA cm⁻² data; two especially well-lit cells have raised the average color of the

50 mA cm⁻² data set to be approximately equivalent to that of 75 mA cm⁻² data set. Nevertheless, the overall trends seem clear.

Footage collected from the side of the electrode demonstrates the differences in bubble attachment in microgravity, where, in the absence of gravity, the difference between the advancing and receding contact angles is minimal and the mean contact angle appears to be smaller. This is shown visually in Fig. 6 and is tabulated in Supplementary Table 1 for a representative sample set of bubbles of approximately equivalent size under 1 g and microgravity. As well as static attachment behavior, the attachment angle of similar sized bubbles highlights the difference in lifecycle, where a bubble that is stable in the growth phase at low gravity may be close to detachment in higher gravity (see Supplementary Note 7). In this case, the mean contact angle in microgravity was found to be approximately 18° smaller than in 1 g. The difference between the advancing and receding contact angles in 1 g increased as bubble size increased. When the maximum difference between the advancing and receding angle is reached as the bubble grows, the bubble can be expected to slide along the electrode surface, potentially coalescing with other bubbles[9]. In the absence of a strong gravitational force, this sliding motion of bubbles does not occur. The observed difference between average contact angles at different gravity levels is not in line with previous work, which found that the contact angles of oxygen bubbles in dilute sulfuric acid are relatively unchanged in microgravity compared with 1 g[31]. A possible reason for this difference is that the electrochemical cells used in the present study have an internal volume two orders of magnitude larger than in this previous study (based on the dimensions given by Matsushima et al.[31]). It follows that the influence of hydrostatic pressure on contact angle in our study would likely be much greater, as hydrostatic pressure is directly proportional to both gravity and the height of the fluid column[32,33]. Our observation that bubble contact angles are affected in microgravity is therefore likely to be more representative of the situation in a practical electrolyzer.

**Comparison and extrapolation of altered-gravity data.** Extrapolation of hypergravity results to reduced-gravity is based on the principal of continuity, where the gravitational field above or below 1 g is continuous and, therefore, physical and biological responses can be expected to also respond with continuity[34]. However, it is important to consider that many phenomena are non-linear as gravity approaches zero and, in general, more variation can be expected between 0 and 1 g compared with between 2 and 3 g, for example[35,36]. This is due to microgravity being an exceptional environment, whereby the absence of key phenomena (e.g. buoyancy or convection) can elicit a unique response. In the present study, the same general logarithmic trend describing the efficiency of oxygen-evolving electrolysis exists between 0.166 and 1 g as seen in hypergravity, which indicates that no significant deviation in the trend should be anticipated across the range from 0.166 g to 8 g. This suggests that, in principle, extrapolation to lunar and Martian gravity levels from hypergravity results is feasible.

The average percentage change relative to 1 g was plotted for each galvanostatic and potentiostatic data set in Fig. 7 across all the g-levels created on the centrifuge in this study. Good agreement between the hypergravity and reduced-gravity data can be seen, particularly for the data sets collected at 75 and 100 mA cm⁻². This is all the more striking when one considers that the reduced-gravity data were collected during parabolic flights, which are subject to a range of external factors such as centrifuge vibration and g-jitter from the aircraft (see Supplementary Fig. 15 and Supplementary Notes 2 and 5), and which

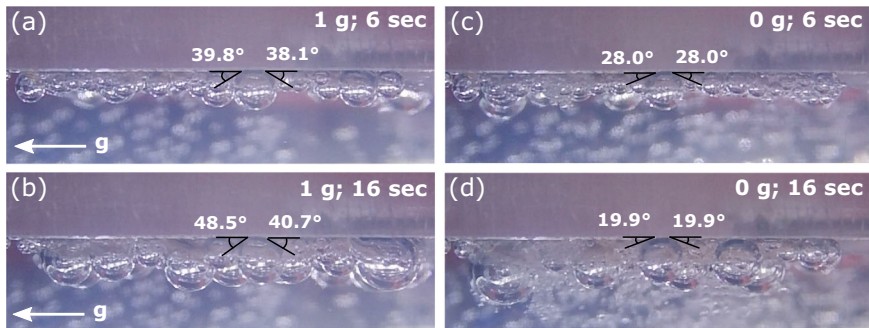

**Fig. 6 Bubble contact angle analysis at 0 and 1 g.** A comparison of bubble attachment angle at 1 g after 6 (**a**) and 16 (**b**) seconds of electrolysis, and at micro-g after 6 (**c**) and 16 (**d**) seconds of electrolysis. Contact angles were measured using imageJ; angles shown are a graphical representation. Average values are shown for each angle and tabulated in Supplementary Table 1.

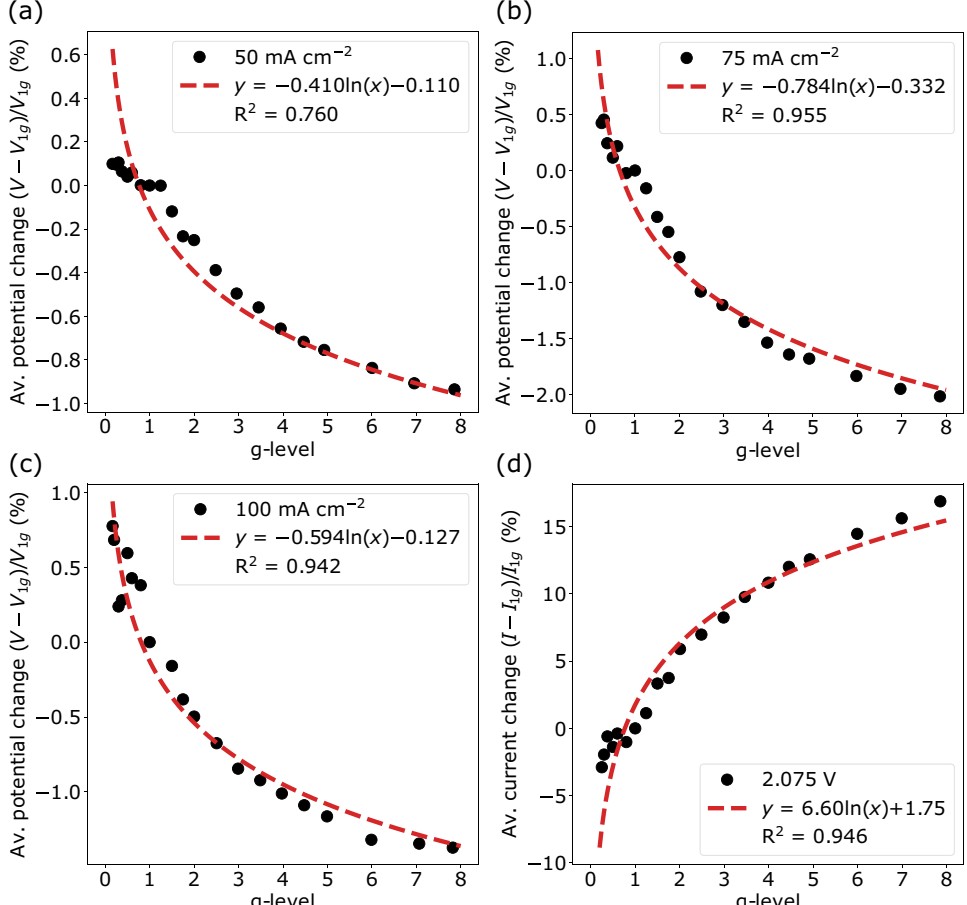

**Fig. 7 Gravity dependent electrolysis efficiency.** The influence of gravity (0.166–8 g) on electrolysis efficiency represented as % change relative to 1 g for datasets collected at **a** 50 mA cm$^{-2}$, **b** 75 mA cm$^{-2}$, **c** 100 mA cm$^{-2}$, and **d** 2.075 V. Source data are provided as a source data file.

can lead to the generation of rather noisy data. Overall, the data in Fig. 7 suggest that hypergravity data sets collected on ground-based centrifuges can be extrapolated to reduced gravity conditions characteristic of the Moon and Mars, which could serve to significantly facilitate the development of electrochemical systems for deployment on these bodies.

**Nonlinear relationship between electrolysis efficiency and gravity.** It was observed throughout the experimental data analysis that electrolysis efficiency decreases nonlinearly as gravity levels decrease. This relationship was found, empirically, to be logarithmic. This section aims to present briefly a physical and

mathematical explanation as to why this nonlinear trend exists. Overpotential has been shown to have a linear dependence on increasing bubble volume on the electrode surface[37]. Given this relationship, the nonlinearity observed must stem from the bubble coverage.

A bubble growing on the surface of an electrode has forces acting upon it, which can either favor or oppose bubble detachment. In the case of a vertical electrode, detachment can result in the bubble sliding or lifting-off (further addressed in Supplementary Note 7). The most significant force pulling a bubble from a surface is buoyancy, which is directly proportional to gravity and the weight of the fluid displaced by the bubble

(i.e., the volume of the bubble multiplied by electrolyte density). Regardless, the bubble will remain attached so long as the forces which oppose detachment are greater than the forces which favor detachment. The force that contributes most significantly to bubble attachment is the interfacial tension force, the force caused by the interfacial tension between the gas, liquid, and the solid electrode. The interfacial tension force ($F_\sigma$) is directly related to the contact diameter ($d_c$) and the bubble contact angle ($\theta$) (Eq. 4). These values are influenced by liquid and electrode properties and are typically investigated experimentally[21].

$$\vec{F_\sigma} = -\pi\, d_c\, \cos(\theta) \tag{4}$$

The nonlinearity of the trends can be related to the comparative rates at which the buoyant and interfacial tension forces change with respect to changing gravity. For example, a bubble of equivalent volume will experience half the buoyant force at 0.5 g as it does at 1 g. While the contact angle and contact diameter will also change based on gravity and its influence on how a bubble sits on a surface (e.g., Supplementary Table 1), this change is comparative, meaning that the interfacial tension force will reduce less than the buoyancy force does. So, in the lower-g condition, interfacial tension becomes more significant than buoyancy. Consequently, as gravity decreases, bubbles are increasingly less likely to depart the surface and can, therefore, grow larger.

While multiple dimensionless quantities have been considered (see Supplementary Note 8), it was determined that the limitations or specificities of each limit their value in the context of this work. The most relevant when characterizing bubble behavior across varying gravitational accelerations is the Bond number[24], which is the ratio of the buoyant force to surface-tension force and captures the two most fundamental competing influences observed in our experiments: surface tension effects (which dominate in low gravity) and buoyant effects (which dominate at higher gravity levels). However, there remain limitations to the Bond number's relevance since it scales linearly with gravity and assumes that the gas bubbles are surrounded by the liquid. A more appropriate dimensionless number, not yet developed, would consider properties of the solid phase, such as surface roughness or surface energy.

As a bubble grows, it can expand along the electrode surface. The maximum contact diameter occurs just before bubble detachment; under low gravity conditions, larger bubbles result in greater contact area and greater interfacial tension force, which can, in turn, offset the increase in buoyant force with increasing volume. Mathematically, as developed by Chesters, the maximum contact diameter ($d_{c\,max}$) is inversely proportional to the square root of the gravitational acceleration (Eq. 5)[38].

$$d_{c\,max} \propto 1/\sqrt{g} \tag{5}$$

When the influence of interfacial tension force increases and the buoyant force decreases as gravity is reduced, the bubble detachment volume increases. As Fritz, Burke and Chesters explain, mathematically, the bubble volume at detachment ($V_{detach}$) has an inverse nonlinear relationship with the gravitational acceleration (Eq. 6)[24,38–40].

$$V_{detach} \propto g^{-1.5} \tag{6}$$

The nonlinear increase in bubble contact area and volume on the electrode surface as gravity decreases can explain the trends observed in this work. While the research described in this paper is purely based on experimental data and analysis, the authors recognize that there have been extensive efforts to model related physical phenomena. Modeling efforts typically make simplifying assumptions, such as focusing on a single bubble or

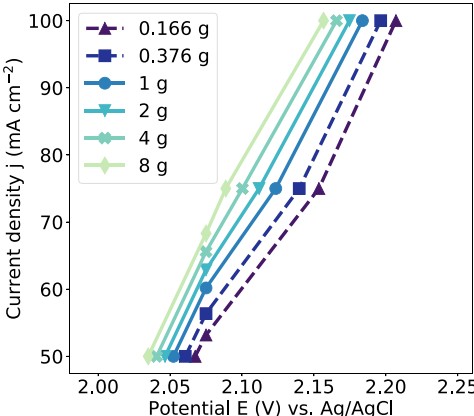

**Fig. 8 Relationship between potential difference and current density.** The current density as a function of potential based on the trends identified for this electrochemical system. Solid lines represent hypergravity data and dashed lines represent reduced-gravity data collected both galvanostatically and potentiostatically. Source data are provided as a source data file.

assuming adiabatic conditions. Nonetheless, robust models have been developed, which agree with the general findings of this research. Modelers such as Kim, Dhruv, Burke and Di Bari have created fluid dynamic models of boiling heat exchangers and single bubble detachment from an orifice[24,41–43]. All found nonlinear, and sometimes discontinuous, trends as gravity scales from microgravity to 1 g. While much work remains to be done to fully characterize fluids in partial gravity, experiments, such as the one described in this paper, are a critical step towards validating the models being developed for reduced-gravity fluid systems.

**Predicting the influence of gravity on gas-evolving electrolysis.** All the data collected throughout this study suggest that oxygen-evolving electrolysis will be less efficient in reduced-gravity; gravity impacts the buoyancy, which in turn alters the bubble growth and departure behavior from an electrode surface. At lower gravity levels, an increased bubble froth layer at the electrode surface will increase ohmic resistance and the anode overpotential, and will thus decrease the efficiency of the system relative to an equivalent system operating in 1 g. While the specific influence of gravity will be highly dependent on the precise nature of the electrochemical system in question, assessing the impact of gravity on the system presented in this work highlights some important general considerations.

The relationship between potential, current density, and gravity found experimentally is shown in Fig. 8, where galvanostatic data points extend horizontally across the graph and the potentiostatic data points extend vertically. As the g-level experienced by the system decreases from 8 g to 0.166 g, a corresponding increase in anode overpotential can be seen, which can be attributed to bubble retention increasing ohmic resistance. The impact of gravity appears to increase with increasing current density, which is consistent with previous work and logically follows from the idea that more bubbles in total would lead to a greater positive or negative influence resulting from their behavior[2,12,26]. However, this trend may only hold up to a certain current density. High bubble production rates in reduced-gravity conditions could work against the effect of reduced-gravity as rapid bubble growth can induce localized convective flow and bubble detachment can generate turbulence in the electrolyte, which in turn can decrease concentration gradients and aid in the detachment of other bubbles[10]. Wang et al.[26] also found that the influence of hypergravity dominated up to a current density of 500 mA cm$^{-2}$,

**Table 1 Practical implication of low gravity electrolysis.**

| Control | Mars | Moon |
|---|---|---|
| Galvanostatic | 0.6% ↑ power | 1.1% ↑ power |
| | Equal product | Equal product |
| Potentiostatic | 6% ↓ power | 11% ↓ power |
| | 6% ↓ product | 11% ↓ product |

The impact on power (W) and production (% oxygen relative to production on Earth at 1 g) of oxygen-evolving electrolysis run with the system investigated in the present study, where the exact control parameter used on Earth is maintained on Mars or the Moon. The current and potential values at 1 g for the highest current density studied (100 mA cm$^{-2}$) have been chosen for this example.

after which the effect of increased bubble production started to counteract the energy saving effect of increased gravity. It is feasible that increasing the current density beyond a certain point could mean that bubble production itself could begin to negate the influence of low-gravity by the same mechanisms.

For a system operating on the Moon or Mars, two key factors need to be considered when assessing the impact of reduced-gravity: the impact on power requirements and the impact on product output. Understanding the relationship between these factors will allow for mitigation and prioritization. Considering the axes of Fig. 8 and the current density range studied for this system, it is evident that a small percentage change in the anode overpotential can generate a large percentage change in the current flow. Current is directly proportional to the charge passed; if it is assumed that there is an equivalent Faradaic efficiency at all g-levels[26], the percentage reduction in current can be assumed to equate to the percentage reduction in oxygen generated by a given system.

Table 1 provides an example of the predicted impact of operating an identical cell with identical parameters on Mars, or the Moon, as compared to Earth. In this case, the impact on electrolysis power requirements would be ~1% when operating on the Moon galvanostatically if an equal product output was targeted. However, the impact of not considering the increased voltage requirement when running potentiostatically would result in 11% less oxygen produced in equivalent time in a system operating using identical parameters to Earth. The precise values given in this example are only relevant to the electrochemical cell used in this study and the chosen current density, but it demonstrates the possible impact of not accounting for the effect of gravity. The expected efficiency or product loss for a larger-scale system could be more pronounced.

While galvanostatic control is most common in industrial-scale systems, there are cases where potentiostatic control may be preferred. Using the oxygen-evolving FFC-Cambridge molten salt process as an example, potentiostatic control is often used to ensure that the potential remains below that which is required to decompose the $CaCl_2$ electrolyte[44]. This could be critical if the level of impurities (such as chlorine) in the oxygen output of a lunar system had to be below a certain level for direct use or further purification[3]. Additionally, if incomplete reduction of an oxide (e.g., lunar regolith) was targeted as the most efficient reduction level for oxygen extraction, potentiostatic control may be more efficient in terms of time and current efficiency. Regardless of whether the system is operated using potentiostatic or galvanostatic control, it will be critical to account for the gravity-induced overpotential at the oxygen-evolving anode that will increase the energy consumption of the system relative to operation on Earth. If overpotential is not accounted for, and the same operating parameters are used, there is the potential to significantly decrease the product output of the system.

The sensitivity of the anodic current density in response to a change in applied potential can inform as to how a given oxygen-evolving system may respond in reduced-gravity environments. The proportional impact of low gravity on product output and power consumption of a specific system can, in fact, be predicted without any altered-gravity experimentation. Looking at Fig. 8 as an example, if we consider the axis in terms of percentage change, the slope of the data obtained at 1 g is approximately 10, as in the current density increases at 10 times the rate of the potential. In turn, this is reflected in the impact on product output in potentiostatic mode being 10 times greater than the impact on power consumption in galvanostatic mode. Assessment of the gradient between potential and current density in any given system would allow for an approximation of the ratio between these efficiency losses in the appropriate current density range. The steeper the gradient is in a system operating in 1 g (i.e., large current change resulting from a small change in applied potential), the greater the possible influence gravity may have on the product output of a system if operated potentiostatically and overpotential is not accounted for. In galvanostatic operation, an increase in energy consumption due to a higher operating potential is inevitable and should be accounted for when designing an oxygen-evolving electrolyzer for the Moon or Mars. Previous studies have found that the overpotential of a gas-evolving electrochemical system was more sensitive to gravity, and comparable to the influence on current density[19]. In such a system, the energy cost resulting from low gravity galvanostatic operation may be more significant. If the power allowance of a gas-evolving electrolysis system was particularly constrained and higher operating potentials were not possible, a lower product output in equivalent time may be preferred.

Determining the exact efficiency losses expected for a system operating on the Moon or Mars would require understanding the expected shift in the polarization curve resulting from changing gravity (i.e., the separation between the lines in Fig. 8). This shift reflects the logarithmic slope of the trendlines identified in Fig. 7, and would most likely need to be determined experimentally as it is highly dependent on the electrolyte, electrode surface, spatial arrangement, and operating parameters. The present work confirms that hypergravity experimentation can be used to identify trends relevant to lunar and Martian gravity.

In this work, the influence of gravity on an oxygen-evolving system was investigated from microgravity ($10^{-2}$ g), through reduced-gravity (including lunar and Martian gravity), to hypergravity (up to 8 g) to try to understand how oxygen-evolving electrolysis systems may behave on the Moon or Mars. Additionally, this study aimed to assess experimentally if hypergravity trends could be extrapolated to lunar gravity, to simplify future research into electrolyzer systems destined for the Moon or Mars.

It was found that the influence of gravity on galvanostatic and potentiostatic electrolysis efficiency in an acidic electrolyte followed a logarithmic relationship across all studied g-levels. Gas bubbles departed the electrode surface more rapidly as the gravitational force, and consequently, the buoyancy increased. The impact of gravity on electrolysis appeared less pronounced in reduced-gravity compared with hypergravity and there was larger variation within data sets. However, in principle there appears to be no reason why logarithmic hypergravity trends could not be extrapolated to lunar gravity (i.e., no drastic change in behavior between 1 and 0.166 g was observed). This result suggests that hypergravity research platforms such as ground-based centrifuges can indeed be used to predict gas-evolving electrolysis efficiency at lunar and Martian gravity, provided that the influence of external factors such as vibration and centrifugal effects are taken into account.

In the present electrochemical system studied between 50 and 100 mA cm$^{-2}$, it was found that the increase in anodic overpotential in reduced-gravity would result in higher power consumption (up to 1.1%) or lower product yields (up to 11%) in a system operating on the Moon or Mars relative to Earth. In general, the effect that gravity had on electrolysis was found to become more pronounced as the current density was increased. The impact on product output in the studied system is more significant due to the relationship between overpotential and current density. This could be mitigated by increasing the operating potential according to the anticipated gravity-induced increase in overpotential, or possibly by introducing countermeasures such as structured electrodes or forced convection. Overall, higher power budgets will likely be needed for Martian and lunar multiphase gas-evolving electrolyzers and this work shows that the impact of gravity on a specific system can be assessed with ground-based hypergravity research.

## Methods

Hyper-gravity experiments from 1 to 8 g were conducted in a laboratory setting using a short-arm centrifuge (radius = 25 cm); two cells on opposite arms were used to collect data. An orbital shaking plate (Orbital Incubator STUART SI 50) operating between 0 and 200 RPM was used to examine the relationship between electrolysis efficiency and vibration/shaking motions in the studied system.

Reduced-gravity electrolysis experiments between 0 and 1 g were carried out on Novespace's Airbus A310 aircraft. Data were collected over three flights consisting of 30 microgravity parabolas each. The microgravity level achieved during parabolic flight is approximately 10$^{-2}$ g. A rotating short-arm centrifuge (the same system as used for the hypergravity experiments) equipped with four electrochemical cells was operated during the microgravity flights to create artificial reduced-gravity; two additional cells were housed separately to the centrifuge and were kept stationary for the duration of the microgravity flights. The minimum duration of reduced gravity was approximately 22 s; as such all experiments were carried out for 18 s to ensure completion prior to the parabola exit maneuver.

**Centrifuge experiment set-up.** The experiment rack was designed to comply with Novespace's safety, interface, and design requirements and is shown in Supplementary Fig. 1 (see also Supplementary Note 1). Four electrolysis cells were attached to a centrifuge (radius = 25 cm), as shown in Supplementary Fig. 2. Baskets were attached to each arm of the centrifuge with low-friction ceramic bearings (SMB Bearings CCZR-696PK) and a stainless steel pin. Baskets were free-swinging in all hypergravity experiments, and locked in place horizontally with 3D-printed PLA (polylactic acid) inserts for all reduced-gravity experiments. The centrifuge table structure was affixed on vibration dampening pads (Ganter GN148-60-M10-A-1-57) to decouple the vibrations of the aircraft from the experiment.

A hollow aluminium shaft was attached to the table structure with two bearings (NSK PSF30CR) and rotated using a bi-directional stepper motor (Oriental motors PK5913HNAW with Oriental motors CVD528B-K motor driver). The motor driver was controlled by a PID (proportional integral derivative) closed feedback loop, utilizing a 3-axis accelerometer (DFRobot SEN0142) positioned at the center of one cell basket to ensure the desired artificial gravity was maintained throughout an experiment. A through-bore, 42-circuit gold-gold contact slip ring (Senring H3099-42S-D-52824) enclosing the shaft fed all power and data cables to the rotating system. All cables were fed through a braided metal shielding sleeve and/or a metal conduit to minimize signal interference. Two 3D-printed PLA camera holders were attached to each basket and contained an action camera (AKASO V50 X), each fitted with a 15× macro lens to observe the front and side of each working electrode. Footage was recorded at 60 frames per second with a resolution of 1080p. A battery powered LED strip was fitted around the basket to illuminate the electrode from all sides. Two additional cells were attached to a stationary table on vibration dampening pads (Ganter GN356-25-20-20-SS-55) next to the centrifuge for microgravity data collection during flight. A Type K thermocouple (RS PRO 397-1488) was attached to the outer wall of one stationary cell.

**Electrochemical cell.** Electrochemical cells (Fig. 9) were custom-designed and machined from polycarbonate, with internal dimensions of 40 × 40 × 28 mm (X × Z × Y; anodic chamber) and 40 × 40 × 21 mm (X × Z × Y; cathodic chamber). The lid was sealed with a silicone gasket and 12 bolts to ensure each cell was liquid and gas-tight. The anodic and cathodic chambers were separated by a 25 × 25 mm Nafion$^{TM}$ N-117 ion exchange membrane (Alfa Aesar, 0.180 mm thick) sandwiched between two polycarbonate windows with two silicone gaskets on either side. The entire assembly was sealed with silicone gel within a groove in the cell walls to limit electrolyte mixing. Vertical polycarbonate electrode holders were suspended from the cell roof parallel to the membrane. The counter electrode was oriented towards the membrane, while the working electrode was oriented towards

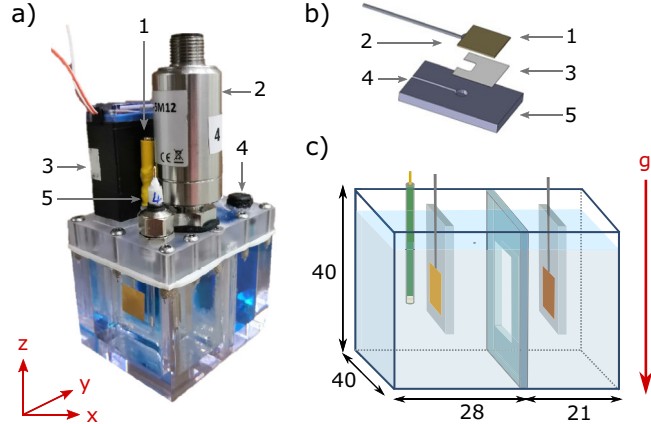

**Fig. 9 Electrolysis cell design. a** The polycarbonate cell showing (1) the electrode connection, (2) pressure sensor, (3) pressure release valve, (4) vent, (5) reference electrode; **b** Electrode preparation method showing: (1) the foil electrode, (2) soldered wire, (3) epoxy glue layer, (4) epoxy-filled groove for wire and solder, and (5) polycarbonate electrode holder; **c** the spatial arrangement of the gold anode, Ag/AgCl reference electrode (shown as green for clarity), Nafion membrane window, and copper cathode. Numbers indicate cell dimensions (in mm).

the cell wall so that the surface of interest could be observed. The anodic chamber lid contained feedthroughs for the working electrode wire (which was subsequently connected to a 2 mm banana socket), an IP68 rated cable gland for the reference electrode, a 0–50 mbar gauge pressure sensor with a ±0.25% full range accuracy (Cynergy3 IPSL-G0050-5M12/PRO), and a direct-acting 2/2-way solenoid pressure release valve (Bürkert 00290108). The cathodic chamber lid was fitted with a counter electrode wire feedthrough connected to a 2 mm banana socket, and a membrane vent rated to 300 mL/min (Amphenol LTW, VENT-PS2NBK-O8001).

**Electrochemical system.** Electrolysis was carried out both galvanostatically and potentiostatically with a three-electrode system to investigate the impact of gravity on an oxygen-evolving working electrode. 5 M sulfuric acid (titration grade, VWR Chemicals), anhydrous copper sulfate (98%, Alfa Aesar), and HPLC grade water were used to prepare both electrolyte solutions (total volume of 69 mL). The cathodic electrolyte (32 mL) consisted of copper sulfate (1.135 M) in dilute sulfuric acid (0.75 M), while the anodic electrolyte (37 mL) was dilute sulfuric acid (0.75 M). The anodic and cathodic reactions are given by Eqs. (7) and (8), respectively. Copper sulfate was chosen for the cathodic electrolyte to suppress hydrogen production on the aircraft (necessary for regulatory reasons); the anodic electrolyte was free from copper sulfate for improved electrode preservation and observation.

$$\text{Anode} : 2H_2O \rightarrow 4H^+ + 4e^- + O_2 \qquad (7)$$

$$\text{Cathode} : Cu^{2+} + 2e^- \rightarrow Cu \qquad (8)$$

Gold foil (0.025 mm thick, Premion 99.985%, Alfa Aesar) cut to 1.25 × 1.25 cm (giving an electrode area of 1.56 cm$^2$) was used as the anode in all experiments, while copper foil (0.025 mm thick, annealed uncoated, 99.8%, Alfa Aesar) of equivalent size was used as the cathode in all experiments. New electrodes were used for each parabolic flight (an experiment set of 30 data points). A tin-coated copper wire was soldered to the back of each electrode and fed through the electrode holders and cell lids. Electrodes were fixed flat on the surface of polycarbonate electrode holders using a two-part epoxy glue (Gorilla Glue); the wire and solder assembly was embedded into an epoxy-filled groove machined into the polycarbonate. The embedded wire was then covered with silicone to ensure that the square electrode was the only electroactive metallic surface. An Ag/AgCl gel electrolyte reference electrode (Pine Research; length: 60 mm, OD: 3.5 mm) designed for aqueous systems and fitted with a ceramic frit was used for all experiments. The electrode design and arrangement are shown in Fig. 9. Electrochemical Impedance Spectroscopy (EIS) was performed under the following experimental parameters: starting frequency = 200 kHz, ending frequency = 100 mHz, DC bias = 0, 0.75, 0.85 V (vs. the open circuit potential), AC excitation amplitude = 10 mV. The series resistance, Rs, was taken as the high frequency intercept on the x-axis of the resulting Nyquist plot. The procedure was carried out on the cells before and after oxygen-producing bulk electrolysis, indicating an initial Rs of ~0.9 Ω (~1.4 Ω cm$^{-2}$) which, post-electrolysis at 1 g, rose to ~1.0 Ω (~1.56 Ω cm$^{-2}$). All potential values throughout this manuscript are reported without iR-compensation.

A time gap of at least three minutes was maintained between all electrochemical experiments to limit the influence of any concentration gradients that may have formed as a result of the previous experiment. The average temperature during all data sets was 21.5 °C, with a maximum variation of 2 °C across any given data set. The average pressure inside the aircraft was 856 mbar; all ground-based hypergravity experiments were conducted at ambient pressure.

**System control and data acquisition**. The electrolysis data were obtained using a Biologic VMP3 16-channel potentiostat. Potentiostat cable extensions on the centrifuge between the slip-ring and cells were comprised of five coaxial cables for the working lead, counter lead, working sense, counter sense, and reference lead, which were woven together and shielded further with an external braided metallic sleeve and insulating plastic sleeve. The cable design emulated that of the cables supplied with the Biologic VMP3. A MyRIO 1900 (National Instruments) microcontroller controlled via LabVIEW software (National Instruments) was used for the centrifuge control and all additional data acquisition. The MyRIO communicated with the potentiostat control computer via a WiFi connection to sync the timestamp of both computers to facilitate data analysis. Thermocouple data were logged using an independent temperature data logger (Omega HH306A). Camera data were stored on internal SD cards. The data acquisition (DAQ) and control architecture of the experiment is shown in Supplementary Fig. 3.

**Data analysis**. The average potential or current was obtained between 4 and 18 s for each galvanostatic or potentiostatic experiment respectively to remove the influence of the bubble nucleation and potential/current stabilization period at the start of the electrolysis. Examples of the raw data sets are shown in Supplementary Figs. 16–19. To enable comparison between data sets from different cells, and across different altered-gravity platforms, the percentage change with respect to 1 g was calculated. To remove the influence of baseline shift in Figs. 3, 4, and to ensure the error bars represent the variation in the trend rather than the baseline, the data displayed are the average of this percentage change applied to the average 1 g measured across the repeats.

The files containing accelerometer and pressure data were trimmed to 18 s based on the timestamp of the corresponding electrolysis file prior to further analysis. The acceleration experienced by the cells in the z-axis was averaged across 18 s to give the mean g-level of a given experiment, which was then compared to the targeted g-level (Supplementary Fig. 4 and Supplementary Fig. 5). The g-level measured by the accelerometer data was equivalent to the g-level at the bottom edge of the square electrode. The gravity gradient across the electrode surface and in the surrounding electrolyte was calculated based on the RPM (revolutions per minute) at a given acceleration, the known distance of the accelerometer from the axis of rotation, and the increased or decreased distance to the axis of rotation at any given point across a 2D surface (Supplementary Fig. 6 and Supplementary Fig. 7). The change in distance was calculated assuming a horizontal basket position in all cases.

To assess the level of vibration during each experiment, a 20-point rolling average was subtracted from the raw acceleration data in all three axes to give a zero baseline. The absolute values of all deviations from this baseline (i.e., vibrations) were plotted and the sum area underneath the curve was calculated to give the sum vibrational intensity of a given axis across each experiment. An example of this procedure is shown in Supplementary Fig. 8. The sum of all three axes was taken to compare the overall motion experienced by a cell across different experiments (Supplementary Fig. 9). The vibrational frequency was computed by counting all peaks during the 18 s experiment using the Python scipy.signal module and then calculating the peaks per second.

Camera footage was trimmed to 20 s from the key frame prior to the first frame showing bubble nucleation. Eleven frames were extracted from each experiment at two second intervals, inclusive of the first and last frame. Each set of frames was scaled and the rotation was adjusted using the known 1.25 cm electrode edge. A 1 cm$^2$ frame was cropped from the center of each electrode for comparison. Approximation of the bubble froth coverage and retention was achieved by converting each frame (excluding 0 and 20 s) to grayscale and calculating the average normalized pixel color across an experiment, with 0 being black and 1 being white. To remove the influence of different lighting baselines the average percentage change relative to the average of 1 g was calculated. To compare the change in bubble attachment angle between 1 g and microgravity, frames corresponding to 6 and 16 s were selected from the side camera footage of a stationary cell and unhindered bubbles ($n = 16$) were selected for analysis. Here, a gas bubble submerged in liquid is investigated and thus the outside angle is of interest. Images were analyzed using the ImageJ Contact Angle plug-in. The advancing and receding attachment angles were measured three times per bubble and the results were averaged. Additional information pertaining to this measurement can be found in Supplementary Note 6.

## Data availability
The source data generated in this study have been deposited in the Enlighten database under accession code https://doi.org/10.5525/gla.researchdata.1210. Source data are provided with this paper.

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

## Acknowledgements

B.A. Lomax thanks ESA, Metalysis Ltd, and the University of Glasgow for funding through the ESA Networking/Partnership Initiative (4000125409/18/NL/MH/mg), and also thanks the UK Space Agency for support. G.H. Just acknowledges the support of the University of Manchester's EPSRC Doctoral Training Partnership, ESA's Network & Partnership Initiative (4000130229/20/NL/MH/hm), the FAIR-SPACE Hub (RN0344) and the Institution of Mechanical Engineers (EAC/KDF/OFFER/20/033). P.J. McHugh thanks the Royal Society for a PhD studentship. P.K. Broadley acknowledges the funding support of the University of Manchester's EPSRC Doctoral Training Partnership. G.C. Hutchings acknowledges the University of Manchester for support through the SEI Internship programme. P.A. Burke extends thanks to the Johns Hopkins University Applied Physics Laboratory for its support. M.J. Roy acknowledges support from the EPSRC (EP/L01680X/1) through the Materials for Demanding Environments Centre for Doctoral Training. M.J. Roy and K.L. Smith acknowledge support from the FAIR-SPACE Hub (RN0344). M.D. Symes thanks the Royal Society for a University Research Fellowship (UF150104) and the EPSRC (EP/K031732/1). The authors acknowledge ESA and Novespace for the opportunity to participate in the 73rd and 74th parabolic flight campaigns. Special thanks must go to Nigel Savage and Emannuelle Auburt (ESA) for all their support. Thibault Paris, Yannick Bailhé, and the whole Novespace team are thanked for all their assistance throughout the campaigns. The authors acknowledge Robert Lindsay, Rafael Leiva-Garcia, Stuart McIntyre and the workshop team from the University of Manchester for all their assistance, and Steven Belcher and the Bindlach group for facilities access.

## Author contributions

B.A.L., G.H.J., P.J.M., P.K.B., and G.C.H. constructed the electrolysis systems, carried out experimental work and took part in the parabolic flight campaign. B.A.L. and G.H.J. analyzed the majority of the data. P.A.B. contributed the discussion on bubble physics. M.J.R., K.L.S., and M.D.S. supervised the project. B.A.L. and M.D.S. wrote the paper.

## Competing interests

The authors declare no competing interests.
