## [Peer Review File · Nature Communications]

Predicting the efficiency of oxygen-evolving electrolysis on the Moon and MarsREVIEWER COMMENTS

Reviewer #1 (Remarks to the Author):

In this study, Lomax et al. characterize the behavior of electrochemically generated O₂ bubbles in different apparent gravitational fields. An electrochemical cell mounted in a centrifuge was used to simulate high gravity fields, whereas reduced simulated gravity was achieved by running the centrifuge cell on a parabolic flight. The electrochemical performance was characterized potentiostatically and galvanostatically over a 4-18 second period. Bubble size, contact angle, and frothiness as a function of gravitational field were characterized using image analysis. By reporting the average change in potential or current, the authors demonstrate an empirical method to predict behavior in microgravity environments using data from supergravity experiments. These results are impactful in that such a scaling analysis could allow for supergravity experiments in a centrifuge to be used as a substitute for the expensive and time constrained methods used for microgravity experiments. Nonetheless, I have a number of comments and minor corrections which should be addressed before consideration for acceptance.

1. Could the authors comment more on the logarithmic relationship between overpotential and gravitational field? They cite previous works which have also identified this logarithmic relationship, but I think the authors should make clear that this is an empirical correlation. Judging by the way the data appears in Figure 10, it almost looks like a Nernstian shift in the polarization curves. I'm guessing that much of this shift is due to an increase ohmic resistance which shifts the apparent overpotential, but it's possible that reactant transport or a change in electrochemically active surface area are also behind this shift. In any case, the authors report a variety of fitted slopes and intercepts from the data sets, but what should the reader be taking away from this? Would it be useful to compile the fitted parameters into a plot or data table?
2. If a theme of this paper is that the high gravity experimental data can help predict trends into low gravity environments, is there a good dimensionless number that could be relied on to describe a transition in the relevant physics governing transport phenomena? The Bond Number comes to mind – I would expect surface tension to be more important in the microgravity regime. Other possible dimensionless numbers which could be relevant are the Grashof number and the Froude number.
3. In Figure 3, what is the ohmic resistance between the reference electrode and the working electrode for this system? Is this potential IR-corrected? It may be interesting to report EIS data before and after bubble formation to get an estimate in the change in ohmic resistance. Given that potential is reported in Figure 10 as versus Ag/AgCl in the axis title, it may be more consistent to also label the potential as relative to Ag/AgCl in Figure 3.
4. Does the dataset in Figure 3 have error bars similar to the data reported in Figure 4? Its possible that perhaps the data in Figure 3 was consistent enough that the error bars are smaller than the data points.

5. How much capacitive charging current do you expect in your experiments? 4-18 seconds is on the shorter end for an experiment, and capacitive charging current may be significant. Could the authors add a few demonstrative chronopotentiometry and chronoamperometry plots to the supporting information to show in general what regime was being averaged over? Even if the amount of capacitive charging current is substantial after 4 seconds, I don't think it undermines the work of the paper, but I think it would be useful information for the readership to have.

6. Have the authors tried using the centrifuge setup to run the electrolyzer at higher current densities or for longer time scales? These would be the operating regimes where gas bubble buildup would have the largest impact on performance. I understand that the parabolic flight experiments are limited by time and possibly current density at the counter electrode where copper deposition had to be carried out instead of HER for safety reasons. If an advantage of using the centrifuge is greater flexibility over your operating parameters, some analysis of the electrolyzer performance in a regime that has substantial gas bubble build up would be beneficial.

7. Figure 9 is mislabeled as Figure 1 (Line 483). For the bottom right panel, if it is a change in current, the axis should read $(I-I_{1g})/I_{1g}$

Reviewer #2 (Remarks to the Author):

This manuscript described the effect of gravity on oxygen evolution reactions. Water electrolysis in a microgravity environment is vital and essential for the future space development, although still it requires highly advanced research. What is the most interesting point is that various microgravity strength could be created by introducing the centrifuge device during a parabolic flight experiment. The unique idea breakthroughs the microgravity acceleration fixed by the facility. Therefore, the authors impressively examined the bubble generation behavior from the micro- to high-gravity levels successfully. They could discover the general relationship between the gravitational acceleration and the oxygen bubble growth, which is discussed very carefully and theoretically.

The following optional comments would help for revising the manuscript.

1. From the experimental results, the empirical formulas were fitted with the gravity acceleration as the parameter. Is it possible to explain the physical meanings of the factor and the intercept?

2. The data used the average values for 4 -18 seconds. However, the actual transient data (Bubble diameter, Potential, etc.) as the supporting information may discuss the additional phenomena (coverage and porosity) for interesting researchers.

3. Figure 9: There is the mistyping in the caption.

At the result of 50 mA cm⁻², the potential change has the discontinuous point around 2 g. Is there any reason for this?

Reviewer #3 (Remarks to the Author):

NCOMMS-21-27729: Predicting the efficiency of oxygen-evolving electrolysis on the Moon and Mars

GENERAL COMMENTS: This manuscript reports an experimental study on the effect of gravitational fields ranged from 0.166 g to 8 g, on the water electrolysis processes. The aim of the study is to assess the energy efficiency of the oxygen production via electrolysis under lower gravitation field, a scenario assuming that the production process is taking place on the Moon or Mars where, in comparing with terrestrial conditions, an issue of reduced g levels needs to be addressed. I must say that I enjoyed going through the manuscript as well as the Supporting Information material on such an interesting topic. The major findings obtained from the study are: a) it is demonstrated the relevance of the decreased electrolysis production rate to the reduced g-levels, and b) categorically the influence of gravity on the oxygen production electrolysis efficiency at the reduced g-level appeared to still follow a logarithmic relationship found in the high-g conditions which can be more readily accessible.

Arguably this study, as suggested in the title, is a long shot project in terms of its overall aim, but by no means its importance should be under-estimated. In contrast the significance is highlighted as its implication/application potentially relevant to the humankind's future. The data showed the influence of gravity on galvanostatic and potentiostatic electrolysis efficiency followed a logarithmic relationship across all studied g-levels, hence people can keep their feet on the earth when investigating the water electrolysis processes under the reduced g-levels. The data generated from the study is also fundamentally important as they could provide a better understanding of g-level's impact on the life cycle and behaviours of bubbles in electrolysis processes. Given the fact that the bubble generation and growth in the electrolysis are similar to that of boiling processes, significance of the results will be appreciated in the communities on the multiphase fluid dynamics, thermodynamics and electrochemical engineering etc.

I recommend the manuscript is accepted for publishing with minor changes (see below the Suggested Improvements).

EXPERIMENTAL ASPECT: The goal of the study was to generate experimental data on how a reduced gravitational field affects water electrolysis process. This goal was achieved by conducting the experiments in both the laboratory (for high-g levels) and on a Novespace's aircraft during parabolic

flights (for low-g levels) using a purpose-built centrifuge electrolyser. The experimental setup including the electrolysis system, instrumentation, data acquisition system and the test procedures appeared well designed and fit the purpose. Checking through the reported data they look technically sound.

SUGGESTED IMPROVEMENTS:

1. The reviewer would appreciate that if the authors can provide their discussion on what physics are behind the logarithmic trend/law.
2. To provide a better vision of the data points against the logarithmic law, I wonder if the plots in Figure 4 should also be presented against log scale of the G-levels, (with 0-g data removed). The new plots can be put side by side with the existing plots.
3. The results of contacting angles (presented in Figure 8 & Table 1) under different g-levels are very interesting. The contacting angle measurement method could be more detailed, and the uncertainty of the measurement should be provided.
4. Line 483: Figure 1 should be Figure 9.

CLARITY AND CONTEXT: the manuscript was excellently structured and well written. The results have been provided with sufficient context and consideration of previous work.

REFERENCES: The manuscript references list is comprehensive and well covered the areas relevant to the research subject.

REVIEWER'S EXPERTISE: Academic. Chemical Engineering background with a research experience in electrolysis process under hi-g. Has expertise in both experimental multiphase fluid dynamics, and process system engineering. Published considerable number of research papers in reputed peer reviewed journals and international conferences.

We thank the referees for their careful reading of our manuscript and for their suggestions for improvement. In the following, we answer each of the referees' comments (in italics) in turn (our answers are in bold typeface). Where we have changed the manuscript or Supplementary Information in response to these comments, we have highlighted these in yellow.

Reviewer #1 (Remarks to the Author):

In this study, Lomax et al. characterize the behavior of electrochemically generated O₂ bubbles in different apparent gravitational fields. An electrochemical cell mounted in a centrifuge was used to simulate high gravity fields, whereas reduced simulated gravity was achieved by running the centrifuge cell on a parabolic flight. The electrochemical performance was characterized potentiostatically and galvanostatically over a 4-18 second period. Bubble size, contact angle, and frothiness as a function of gravitational field were characterized using image analysis. By reporting the average change in potential or current, the authors demonstrate an empirical method to predict behavior in microgravity environments using data from supergravity experiments. These results are impactful in that such a scaling analysis could allow for supergravity experiments in a centrifuge to be used as a substitute for the expensive and time constrained methods used for microgravity experiments. Nonetheless, I have a number of comments and minor corrections which should be addressed before consideration for acceptance.

1. Could the authors comment more on the logarithmic relationship between overpotential and gravitational field? They cite previous works which have also identified this logarithmic relationship, but I think the authors should make clear that this is an empirical correlation. Judging by the way the data appears in Figure 10, it almost looks like a Nernstian shift in the polarization curves. I'm guessing that much of this shift is due to an increase ohmic resistance which shifts the apparent overpotential, but it's possible that reactant transport or a change in electrochemically active surface area are also behind this shift. In any case, the authors report a variety of fitted slopes and intercepts from the data sets, but what should the reader be taking away from this? Would it be useful to compile the fitted parameters into a plot or data table?

We thank the reviewer for this suggestion to improve the manuscript and agree that a discussion into the physical and mathematical explanation of the trend identified adds value to the paper. As such we have included a new section titled "Nonlinear relationship between electrolysis efficiency and gravity". Furthermore, we clarify that the relationships identified are empirical and make reference to relevant modelling work.

The reviewer's interpretation of the data in Figure 10 agrees with our understanding (see lines 63 – 73 where bubble impact on ohmic resistance, electrocatalytic surface, and concentration gradients are discussed). The key take-away from these empirically determined relationships is that, with the polarisation curve at 1 g and the slope ($m \cdot \ln(x)$) identified through hypergravity experimentation, it is possible to determine how strongly a system is impacted by gravity at a given current density, and to predict the operating efficiency in lunar or Martian gravity conditions. This explanation has been expanded upon in the final paragraphs in the discussion, where the main take-aways of the paper have been clarified. Furthermore, we thank the reviewer for the suggestion to collate the fitted parameters into a table. While the trends identified and discussed are interesting in a general sense of demonstrating these relationships (gravity dependent bubble behaviour and electrolysis efficiency), the slope and intercept should be determined experimentally for a specific system (as is

discussed in text) and so focusing on and tabulating the values identified for the system used in this work may take away from the primary message of the manuscript.

2. If a theme of this paper is that the high gravity experimental data can help predict trends into low gravity environments, is there a good dimensionless number that could be relied on to describe a transition in the relevant physics governing transport phenomena? The Bond Number comes to mind – I would expect surface tension to be more important in the microgravity regime. Other possible dimensionless numbers which could be relevant are the Grashof number and the Froude number.

We carefully considered the application of multiple dimensionless numbers to this work. However, it was determined that the limitations or specificities of each number limits the value of their use in this particular research; the reasoning in each case is detailed below. Regardless, we thank the reviewer for this interesting suggestion.

Past work has attempted to use various dimensionless quantities to scale bubble behaviour across gravitational acceleration levels. However, the bubble behaviour observed in the reduced gravity experiments cannot be fully accounted for with traditional scaling techniques. Pamperin, for example, used the Weber number to study bubble detachment from a submerged orifice in reduced gravity [1]. The Weber number is the ratio of inertial forces to cohesion forces (surface tension) acting on a multiphase flow (Eqn. 1) [1]. Although surface tension forces are expected to be significant in the reduced-gravity experiments described within this paper, the Weber number is not germane to the work described herein. The Weber number does not account for the change in gravitational acceleration. Secondly, bubbles nucleating, growing, and detaching through the process of electrolysis have minimal inertial forces acting upon them. Pamperin's experiments, by contrast, studied bubble detachment via gas jetting through an orifice [1].

$$We = \frac{\rho v^2 l}{\sigma} \quad (1)$$

The Grashof number is also mentioned by the reviewer as a candidate dimensionless quantity to scale fluid flows across gravity levels. The Grashof number is a ratio of buoyant to viscous forces acting on a fluid flow (Eqn. 2) [2]. While it is relevant to buoyant flows, the Grashof number is typically used to study single-phase flows experiencing natural convection, caused by temperature gradients. In contrast, the experiment conducted for this research aimed to maintain constant temperatures. The buoyant flows, instead, were caused by the electrolytic nucleation of gas bubbles. Lastly, the research conducted also did not study the influence of viscosity (the second main term in the Grashof number) on bubble nucleation and growth.

$$Gr = \frac{g \beta \Delta T l^3}{\nu^2} \quad (2)$$

The Froude Number is another dimensionless quantity frequently used to characterize the influence of gravity on a fluid flow. The Froude number is the ratio of inertial forces to external body forces, often simply defined as the body force due to gravity (Eqn. 3) [3]. It is interesting to note that the Froude number scales nonlinearly with gravity, similar to the nonlinear relationship found in the research discussed in the paper. However, similarly to the Weber number, the Froude number's

main parameter focuses on a flow's inertial forces. Since electrolytic bubble nucleation and growth occur in a non-flowing liquid, the Froude number does not completely apply to our work.

$$Fr = \frac{u}{\sqrt{gl}} \quad (3)$$

Finally, the Bond Number is perhaps the most frequently used dimensionless quantity when attempting to characterize bubble and droplet behaviour, especially bubble shape. The Bond Number is the ratio of gravitational to surface tension forces (Eqn. 4) [4]. The Bond Number appears very applicable to the problem being studied: accounting for both the buoyant and surface-tension forces acting on the bubbles. However, there remain some important limitations to the use of the Bond Number. First, the Bond Number makes the assumption that the bubble is completely surrounded by liquid. That is, the Bond Number fails to account for any solid-fluid interactions. It has been shown that the properties of the solid, on which the bubble is adhered, can greatly influence the detachment time and volume of the bubble [5-9]. Since electrolysis efficiency is directly related to the release of bubbles from the electrode, the properties of the solid electrode, such as surface roughness or surface energy, must be accounted for in any dimensionless quantity used to scale across gravity levels. Finally, the Bond Number suggests a linear relationship as bubble behaviour is scaled across gravity levels. The experimental trends presented in this research counter this, by displaying a nonlinear, logarithmic relationship between electrolysis efficiency and gravity level.

$$Bo = \frac{\Delta\rho g l^2}{\gamma} \quad (4)$$

- [1] Pamperin, O., and Rath, H., "Influence of buoyancy on bubble formation at submerged orifices," *Chemical Engineering Science*, Vol. 50, No. 19, 1995, pp. 3009-3024.
- [2] Bergman, T.L., Lavine, A.S., Incropera, F.P., "Fundamentals of Heat and Mass Transfer," Wiley, Somerset, 2011, pp. 408.
- [3] White, F.M., and Xue, H., "Fluid mechanics," McGraw-Hill, New York, NY, 2021, pp. 294.
- [4] Clift, R., Grace, J.R., and Weber, M.E., "Bubbles, drops, and particles," Acad. Pr, New York, NY [u.a.], 1978, pp. 26.
- [5] Burke, P.A., and Dunbar, B.J., "Development of Computational Fluid Dynamic (CFD) Models of the Formation and Buoyancy-Driven Detachment of Bubbles in Variable Gravity Environments," American Institute of Aeronautics and Astronautics, 2021,
- [6] Sakuma, G., Fukunaka, Y. & Matsushima, H. Nucleation and growth of electrolytic gas bubbles under microgravity. *Int. J. Hydrogen Energy* 39, 7638–7645 (2014).
- [7] Brinkert, K. et al. Efficient solar hydrogen generation in microgravity environment. *Nat. Commun.* 9, 2527 (2018).
- [8] Xu, W., Lu, Z., Sun, X., Jiang, L. & Duan, X. Superwetting Electrodes for Gas-Involving Electrocatalysis. *Acc. Chem. Res.* 51, 1590–1598 (2018).

[9] Zhao, X., Ren, H. & Luo, L. Gas Bubbles in Electrochemical Gas Evolution Reactions. *Langmuir* 35, 5392–5408 (2019).

3. In Figure 3, what is the ohmic resistance between the reference electrode and the working electrode for this system? Is this potential IR-corrected? It may be interesting to report EIS data before and after bubble formation to get an estimate in the change in ohmic resistance. Given that potential is reported in Figure 10 as versus Ag/AgCl in the axis title, it may be more consistent to also label the potential as relative to Ag/AgCl in Figure 3.

EIS measurements have been carried out on the cells, and the experimental section of the paper has been updated to report the series resistance pre- and post- electrolytic bubble formation. The potential graphs reported are not iR-compensated. Though doing so would change the E1 value in equation 3 for each trend line, the general logarithmic trends would still hold. In terms of our extrapolations discussed in section 3.3, these are reported as % change relative to 1 g, and so iR-compensation should not have any influence. However, this has been clarified in the experimental section for the readers' information. The difference in ohmic resistance measured in our cells before and after electrolysis was $\sim 0.1 \Omega$, at 1 g. Hence the bulk electrolyses had a negligible effect on the series resistances within our cell, aside from the impact of bubble behaviour under different gravity conditions, which is reflected in the trends identified.

We thank the reviewer for this suggestion to improve consistency, Figure 3 and 4 have been updated to clarify that the potential values are vs. Ag/AgCl.

4. Does the dataset in Figure 3 have error bars similar to the data reported in Figure 4? Its possible that perhaps the data in Figure 3 was consistent enough that the error bars are smaller than the data points.

The data in Figure 3 was the average of two repeats that followed a very close trend with one another, however, due to shifted baselines resulting from inherent differences in the cell and electrode, error bars on this average would reflect only the difference in baseline rather than difference in trend (as discussed in the manuscript). Following this recommendation to include the error bars, we have reconsidered presenting the simple average of the repeats and have instead analysed the hypergravity data using the same baseline independent method as the reduced gravity data (i.e., percentage change relative to the 1 g data point calculated for each repeat, averaged for each g-level, and then applied to the average of the 1 g experiment). With this method the baseline influence is removed, and the error bars represent the variation in the trend. This has been clarified in the "Data Analysis" sub-section of the experimental section. The relevant graphs, with error bars, and corresponding text have been updated accordingly to reflect this improved trend assessment of the hypergravity data. As such, the trend lines identified have slightly changed but this does not impact the conclusions in any way.

5. How much capacitive charging current do you expect in your experiments? 4-18 seconds is on the shorter end for an experiment, and capacitive charging current may be significant. Could the authors add a few demonstrative chronopotentiometry and chronoamperometry plots to the supporting information to show in general what regime was being averaged over? Even if the amount of capacitive

charging current is substantial after 4 seconds, I don't think it undermines the work of the paper, but I think it would be useful information for the readership to have.

An example of a raw data set collected at each parameter (chronopotentiometry and chronoamperometry) has been added to the supplementary information Section S.9. As can be seen from these examples, capacitive charging only had an impact on the data within the first 0.5-1 second. Therefore, we believe that using the average of 4-18 seconds (i.e., excluding the start of the electrolysis from the calculated averages), was reasonable to ensure that the data had stabilised, and variation seen could then be attributed to the phenomena of interest.

6. Have the authors tried using the centrifuge setup to run the electrolyzer at higher current densities or for longer time scales? These would be the operating regimes where gas bubble buildup would have the largest impact on performance. I understand that the parabolic flight experiments are limited by time and possibly current density at the counter electrode where copper deposition had to be carried out instead of HER for safety reasons. If an advantage of using the centrifuge is greater flexibility over your operating parameters, some analysis of the electrolyzer performance in a regime that has substantial gas bubble build up would be beneficial.

The authors thank the reviewer for the interesting suggestion; however, the recommended experiments were not possible within the frame of this work. As the reviewer correctly states, the parabolic flight experiments were limited in both time (duration of one parabola) and current density (safety limitation to minimise gas production). As the core purpose of the hypergravity experiments with the centrifuge was a direct comparison to the data collected during flight, we did not carry out experiments under different conditions as part of this study. Additionally, as the experiment rack containing the centrifuge has since been repurposed for another parabolic flight experiment, collection of additional data is not feasible at this time. The investigation of higher current densities in electrolysis systems more representative of technology intended for lunar or Martian operation (i.e., a larger scale electrolyser operating for meaningful durations under hypergravity) is the intended subject of future work that will utilise a large-diameter centrifuge with far fewer experimental limitations and so that work will address this interesting suggestion.

7. Figure 9 is mislabeled as Figure 1 (Line 483). For the bottom right panel, if it is a change in current, the axis should read $(I-I_{1g})/I_{1g}$

We thank the reviewer for pointing out these typos, which we have corrected.

Reviewer #2 (Remarks to the Author):

This manuscript described the effect of gravity on oxygen evolution reactions. Water electrolysis in a microgravity environment is vital and essential for the future space development, although still it requires highly advanced research. What is the most interesting point is that various microgravity strength could be created by introducing the centrifuge device during a parabolic flight experiment. The unique idea breakthroughs the microgravity acceleration fixed by the facility. Therefore, the authors impressively examined the bubble generation behavior from the micro- to high-gravity levels successfully. They could discover the general relationship between the gravitational acceleration and the oxygen bubble growth, which is discussed very carefully and theoretically.

The following optional comments would help for revising the manuscript.

1. From the experimental results, the empirical formulas were fitted with the gravity acceleration as the parameter. Is it possible to explain the physical meanings of the factor and the intercept?

We thank the reviewer for this suggestion to improve the manuscript and agree that a discussion into the physical and mathematical explanation of the trend identified adds value to the paper. As such we have included a new section titled “Nonlinear relationship between electrolysis efficiency and gravity” (see also our answer to Referee 1’s comment 1 above). Additionally, the definitions of the factor and intercept of this relationship are given in relation to Eq. (3).

2. The data used the average values for 4 -18 seconds. However, the actual transient data (Bubble diameter, Potential, etc.) as the supporting information may discuss the additional phenomena (coverage and porosity) for interesting researchers.

Examples of the raw data collected at each parameter (chronopotentiometry and chronoamperometry) has been added to the supplementary information Section S.9 for interested readers and to visually show why the 4-18 second average was taken to avoid the initial portion of the experiment where capacitive charging and/or nucleation may influence averages. As is discussed in the manuscript, quantitative analysis of the transient bubble diameter was not possible within the scope of this work due to the fizzy nature of the bubbles. Rather, the bubble coverage has been estimated using pixel coverage as we believe that this captures the changing fizz layer well. For interested readers, examples of the raw bubble frames used for this calculation are given in Supporting Information Section S.7.

3. Figure 9: There is the mistyping in the caption. At the result of 50 mA cm⁻², the potential change has the discontinuous point around 2 g. Is there any reason for this?

We thank the reviewer for pointing out this typo, which we have corrected.

The discontinuity in the 50 mA cm⁻² data set in Figure 9 reflects the difference between the data collected in hypergravity conditions in a laboratory setting and the data collected in reduced-gravity conditions during parabolic flight. Sources for deviations in the data set (such as g-jitter and mechanical vibration) have been discussed in the manuscript and supporting information. Regardless, we do not believe this deviation in the trends impacts the conclusion that the same logarithmic trend exists in both cases and that, under steady conditions, a continuous trend would be expected above and below 1 g, thus demonstrating experimentally that hypergravity data could be used to predict reduced-gravity behaviour.

Reviewer #3 (Remarks to the Author):

NCOMMS-21-27729: Predicting the efficiency of oxygen-evolving electrolysis on the Moon and Mars

GENERAL COMMENTS: This manuscript reports an experimental study on the effect of gravitational fields ranged from 0.166 g to 8 g, on the water electrolysis processes. The aim of the study is to assess the energy efficiency of the oxygen production via electrolysis under lower gravitation field, a scenario assuming that the production process is taking place on the Moon or Mars where, in comparing with terrestrial conditions, an issue of reduced g levels needs to be addressed. I must say that I enjoyed going through the manuscript as well as the Supporting Information material on such an interesting topic. The major findings obtained from the study are: a) it is demonstrated the relevance of the decreased electrolysis production rate to the reduced g-levels, and b) categorically the influence of gravity on the oxygen production electrolysis efficiency at the reduced g-level appeared to still follow a logarithmic relationship found in the high-g conditions which can be more readily accessible.

Arguably this study, as suggested in the title, is a long shot project in terms of its overall aim, but by no means its importance should be under-estimated. In contrast the significance is highlighted as its implication/application potentially relevant to the humankind's future. The data showed the influence of gravity on galvanostatic and potentiostatic electrolysis efficiency followed a logarithmic relationship across all studied g-levels, hence people can keep their feet on the earth when investigating the water electrolysis processes under the reduced g-levels. The data generated from the study is also fundamentally important as they could provide a better understanding of g-level's impact on the life cycle and behaviours of bubbles in electrolysis processes. Given the fact that the bubble generation and growth in the electrolysis are similar to that of boiling processes, significance of the results will be appreciated in the communities on the multiphase fluid dynamics, thermodynamics and electrochemical engineering etc.

I recommend the manuscript is accepted for publishing with minor changes (see below the Suggested Improvements).

EXPERIMENTAL ASPECT: The goal of the study was to generate experimental data on how a reduced gravitational field affects water electrolysis process. This goal was achieved by conducting the experiments in both the laboratory (for high-g levels) and on a Novespace's aircraft during parabolic flights (for low-g levels) using a purpose-built centrifuge electrolyser. The experimental setup including the electrolysis system, instrumentation, data acquisition system and the test procedures appeared well designed and fit the purpose. Checking through the reported data they look technically sound.

SUGGESTED IMPROVEMENTS:

1. The reviewer would appreciate that if the authors can provide their discussion on what physics are behind the logarithmic trend/law.

We thank the reviewer for this suggestion to improve the manuscript and agree that a discussion into the physical and mathematical explanation of the trend identified adds value to the paper. As such we have included a new section titled "Nonlinear relationship between electrolysis efficiency and gravity" (see also our answer to Referee 1's point 1 above).

2. To provide a better vision of the data points against the logarithmic law, I wonder if the plots in Figure 4 should also be presented against log scale of the G-levels, (with 0-g data removed). The new plots can be put side by side with the existing plots.

The Authors thank the reviewer for the suggestion of this alternative way to visualise the data in Figure 4; logarithmic graphs corresponding to Figure 3 and 4 have been added to the supporting information (new Section S10) due to limited figure space in the manuscript.

3. The results of contacting angles (presented in Figure 8 & Table 1) under different g-levels are very interesting. The contacting angle measurement method could be more detailed, and the uncertainty of the measurement should be provided.

We agree with the reviewer that the results are interesting and have therefore repeated the analysis with a larger subset of bubbles ($n = 16$) to further support the conclusions. The averages for the advancing and receding contact angles, and corresponding standard errors, have been updated in Table 1. Additionally, a section has been added to the supporting information (section S11) that provides additional details regarding the analysis methodology. While highly interesting, contact angle analysis beyond that which is presented in the revised manuscript is beyond the scope of the current work and would benefit greatly from single bubble experimentation. In the present work, the number of visually unhindered bubbles for analysis was limited due to the depth of the electrode when viewed from the side.

4. Line 483: Figure 1 should be Figure 9.

We thank the reviewer for pointing out this typo, which we have corrected.

CLARITY AND CONTEXT: the manuscript was excellently structured and well written. The results have been provided with sufficient context and consideration of previous work.

REFERENCES: The manuscript references list is comprehensive and well covered the areas relevant to the research subject.

REVIEWER'S EXPERTISE: Academic. Chemical Engineering background with a research experience in electrolysis process under hi-g. Has expertise in both experimental multiphase fluid dynamics, and process system engineering. Published considerable number of research papers in reputed peer reviewed journals and international conferences.

REVIEWER COMMENTS

Reviewer #1 (Remarks to the Author):

I appreciate the authors responses to my comments and the thought and consideration that went into addressing different dimensionless numbers which could be relevant to their system. My assessment is that the data and experiments are sound and impactful, but I still have reservations about the analysis and interpretation of the data before I can recommend for acceptance.

I support the addition of a section that specifically discusses the nonlinear relationship between electrolysis efficiency and gravity. Clearly, by changing the buoyancy force, there is a shift in the equilibrium force balance on the bubbles. The authors point out that bubble departure diameters also have a nonlinear dependence on gravity, and the dynamics of bubble departure must affect the total bubble coverage on the electrode surface.

However, I still assert that this balance between surface tension and gravity implies the existence of a dimensionless number which should at least begin to quantify the relative contributions of each force. Based on the authors' discussion in the rebuttal letter, the Bond number seems most relevant. Again, I appreciate this discussion was included in the rebuttal letter, but nothing related to a dimensionless number made it into the updated manuscript. The closest the authors get is in line 539: "In the lower-g condition, interfacial tension >> buoyancy." I think its an interesting point that a good dimensionless number should also include more information about the solid phase. I'm not suggesting the authors need to specify what exactly this should be, nor am I suggesting any new modeling work or new experiments need to be done for this manuscript, but if the authors believe the Bond number is not relevant to their system, they should state so in the main text.

The author's willingness to share more of the raw data is also appreciated, and I agree with their assessment that the capacitive charging was not relevant over the time range that they averaged over. I also agree with the authors choice not to IR-compensate the data, especially since it is my understanding that the change in ohmic resistance over the duration of the experiment can be significant.

For example, in lines 223-225, the authors measure a 0.1 ohm increase in the total cell resistance after the electrolysis experiment at 1 g and claim that this change is insignificant. I would argue the contrary. In Figure S18 you show that for an applied current density of 100 mA cm⁻², the increase in overpotential during the experiment is ~25 mV, just by eye-balling the values against the axis. Comparatively, the increase in voltage due to the change in ohmic drop would be $\Delta V = I \Delta R = (1.56 \text{ cm}^2) * (100 \text{ mA cm}^{-2}) * (0.1 \Omega) = 15.6 \text{ mV}$. Thus, at least 50% of the increase in applied potential can be attributed to the ohmic

resistance of gas bubbles in your system. These points raised in this paragraph need not be addressed in any revisions, but I only bring them up to say that I think there is a lot of information in your raw data which gives insights into the ways in which gas bubble coverage affects the efficiency of water electrolysis. If you have the EIS data for the before/after ohmic resistance in the other gravitational experiments to go in the supporting information, I think the readership would appreciate it.

Reviewer #3 (Remarks to the Author):

The authors have made good revision to the manuscript, and my questions have been addressed in the rebuttal and revised manuscript. I am happy to recommend the paper to move forward for publishing.

We thank the referee for his/her additional comments on our manuscript and for the suggestions for improvement. In the following, we answer each of these comments (in italics) in turn (our answers are in bold typeface). Where we have changed the manuscript or Supplementary Information in response to this second round of comments, we have highlighted these in green (to distinguish them from the changes made in the first round of reviewing, which are shown in yellow).

I appreciate the authors responses to my comments and the thought and consideration that went into addressing different dimensionless numbers which could be relevant to their system. My assessment is that the data and experiments are sound and impactful, but I still have reservations about the analysis and interpretation of the data before I can recommend for acceptance. I support the addition of a section that specifically discusses the nonlinear relationship between electrolysis efficiency and gravity. Clearly, by changing the buoyancy force, there is a shift in the equilibrium force balance on the bubbles. The authors point out that bubble departure diameters also have a nonlinear dependence on gravity, and the dynamics of bubble departure must affect the total bubble coverage on the electrode surface.

We thank the reviewer for supporting the additional section describing the nonlinear relationship.

However, I still assert that this balance between surface tension and gravity implies the existence of a dimensionless number which should at least begin to quantify the relative contributions of each force. Based on the authors' discussion in the rebuttal letter, the Bond number seems most relevant. Again, I appreciate this discussion was included in the rebuttal letter, but nothing related to a dimensionless number made it into the updated manuscript. The closest the authors get is in line 539: "In the lower-g condition, interfacial tension \gg buoyancy." I think its an interesting point that a good dimensionless number should also include more information about the solid phase. I'm not suggesting the authors need to specify what exactly this should be, nor am I suggesting any new modeling work or new experiments need to be done for this manuscript, but if the authors believe the Bond number is not relevant to their system, they should state so in the main text.

We appreciate and agree with reviewer #1's suggestion that a discussion regarding dimensionless numbers should be included in the main text. Dimensionless quantities, which are used to analyze fluid behavior (in this case, scaling bubble behavior across gravity regimes) are powerful tools in fluid mechanics. As a result of this suggestion, we have discussed the most relevant dimensionless number (the Bond Number) in the main text and have included all other dimensionless number discussions in the Supporting Information.

Within the main text's discussion of the Bond number, we have highlighted that the Bond number does in fact quantify the competing influences of buoyancy and surface tension as gravitational acceleration scales from reduced gravity to hypergravity. Again, taking the reviewer's suggestion, we also noted the limitations of the Bond number within the main text, i.e., the Bond number scales linearly with gravity and fails to consider any properties of the solid phase. We agree that a more relevant dimensionless quantity (which has not been developed, as evident via a thorough literature search) would account for solid properties, such as surface roughness and surface energy. We think that this is certainly of interest

for future modelling work and appreciate that the reviewer agrees that this is outside the scope of the current manuscript.

We thank the reviewer for their suggestions and appreciate the discussion about the use of dimensionless numbers.

The author's willingness to share more of the raw data is also appreciated, and I agree with their assessment that the capacitive charging was not relevant over the time range that they averaged over.

We thank the reviewer for considering the additional data and agreeing with our assessment.

*I also agree with the authors choice not to IR-compensate the data, especially since it is my understanding that the change in ohmic resistance over the duration of the experiment can be significant. For example, in lines 223-225, the authors measure a 0.1 ohm increase in the total cell resistance after the electrolysis experiment at 1 g and claim that this change is insignificant. I would argue the contrary. In Figure S18 you show that for an applied current density of 100 mA cm⁻², the increase in overpotential during the experiment is ~25 mV, just by eye-balling the values against the axis. Comparatively, the increase in voltage due to the change in ohmic drop would be $\Delta V = I\Delta R = (1.56 \text{ cm}^2) * (100 \text{ mA cm}^{-2}) * (0.1 \Omega) = 15.6 \text{ mV}$. Thus, at least 50% of the increase in applied potential can be attributed to the ohmic resistance of gas bubbles in your system. These points raised in this paragraph need not be addressed in any revisions, but I only bring them up to say that I think there is a lot of information in your raw data which gives insights into the ways in which gas bubble coverage affects the efficiency of water electrolysis. If you have the EIS data for the before/after ohmic resistance in the other gravitational experiments to go in the supporting information, I think the readership would appreciate it.*

We did not collect EIS data at other gravity levels and unfortunately, additional experimental access to the altered gravity platforms used in this work is no longer possible. However, we believe that the results presented for 1 g, together with the discussion regarding the impact of additional bubble coverage at low gravity on ohmic resistance, still allows the readership to draw insight regarding how the bubble coverage impacts series resistance and electrolysis efficiency.

REVIEWERS' COMMENTS

Reviewer #1 (Remarks to the Author):

The authors have sufficiently answered my questions and addressed my concerns in the updated manuscript. I recommend it for publication. It's a nice paper.